# Responding to Maternal Loss: A Phenomenological Study of Older Orphans in Youth-Headed Households in Impoverished Areas of South Africa

**DOI:** 10.3390/healthcare8030259

**Published:** 2020-08-10

**Authors:** Busisiwe Ntuli, Ephodia Sebola, Sphiwe Madiba

**Affiliations:** Department of Public Health, School of Health Care Sciences, Sefako Makgatho Health Sciences University, P.O. Box 215, Medunsa 0204, Pretoria, South Africa; Ephodia.K.Sebola@pfizer.com (E.S.); Sphiwe.madiba@smu.ac.za (S.M.)

**Keywords:** bereavement, grief, orphans, youth-headed households, South Africa, coping strategies, informal settlements, psychological distress

## Abstract

The aim of this study was to explore how older orphans in youth-headed households (YHHs) experience and respond to maternal death and to examine the strategies they employ to care for their younger siblings. We interviewed 18 older orphans who were purposively selected from YHHs located in informal settlements in the City of Tshwane, South Africa. After the death of their mothers, the orphans lost the family home, lost support from their relatives, lost friendships, lost educational opportunities, and lost childhood. The orphans experienced prolonged pain, sadness, anxieties, fear, loneliness despondency, and deep-rooted and persistent anger towards their mother for dying. They suffered from prolonged bereavement because they had been denied the opportunity to mourn the loss of their parents and yearned persistently but silently for their dead mothers. Dropping out of school to seek employment in order to care for their siblings was one of the main coping strategies that older orphans used. However, dropping out of school early robbed them of their future goals of getting an educational qualification. The orphans had not been prepared for taking on an adult role and were given no support or counselling to help them recover from their parents’ death. Continuous grief counselling should form an integral component of the psychosocial support services that are provided to orphans soon after the death of a parent.

## 1. Introduction

The loss of a mother through death is multifaceted and has multiple potential negative effects for children [1,2], but these effects may not manifest until many months or years afterwards [3]. Research suggests that the loss of a parent to an AIDS-related illness carries more risk factors than any other type of bereavement [2,4]. Such an effect may be psychological, behavioral, educational, economic, and damaging to the child’s health and may lead to uncertainty about their future [2,5,6,7,8]. 

The landscape of orphan care has changed, particularly in countries experiencing a high prevalence of HIV/AIDS, high levels of poverty, and HIV-related stigma [9,10]. As a consequence, children orphaned and rendered vulnerable by HIV and AIDS are at high risk of living in child-headed households (CHHs), a phenomenon that has emerged with the scourge of the HIV/AIDS epidemic in many societies [11]. Child-headed families have become a reality and a permanent feature of society [10], even though this occurrence is in contrast to the African societal norms that expect the extended family to take over the responsibility of taking care of children in the event of parental death [11,12,13]. The phenomenon is more noticeable following the loss of multiple family members, particularly the grandmother who is often a substitute mother. Thus, maternal grandparents in many parts of sub Saharan Africa continue to care for orphans [12]. 

While a CHH is defined as a family unit in which the oldest person, who heads the household, is under the age of eighteen [14], as the CHH phenomenon matures, the head of the household becomes older and this leads to youth-headed households (YHHs). In these households, older siblings take over the role of the caregivers of their younger siblings [11,15]. After the death of the parent/s, the older siblings establish their own households and receive no support from their extended families [9]. Household heads have an enormous task to fulfil in taking care of younger siblings. Heading a household for these young people is associated with psychological and emotional trauma, as well as the social distress associated with the loss of a parent [1,9].

Research has established that HIV orphans in YHHs suffer from psychosocial and economic challenges that affect their ability to cope with life in the absence of adult caregivers [6]. They show high levels of long-term emotional problems and a higher prevalence of depression than other groups [16,17]. Compared to the younger siblings belonging to these households, the heads of these households experience higher levels of psychological and emotional strain. They have higher trauma scores and more signs of psychosocial distress, such as feelings of sadness, loneliness, hopelessness, and worry. They also show higher levels of anxiety, depression, and posttraumatic stress disorder than their younger siblings [2,10,18,19,20]. Having to assume a parental role unprepared and unaided puts an emotional strain on the heads of the households and has a negative impact on their future success.

Researchers suggest that in spite of all the psychosocial challenges that heads of households experience and the fact that they take on adult roles with minimal resources for survival [21], they develop resilience [22,23] and can have better outcomes. Popoola and Mchunu [24] found some level of coping with the loss of the parents and their parenting roles. The coping strategies they employ could be either negative or positive. Some heads of households may resort to transactional sex in order to survive [25] and others may drop out of school in order to find employment [5,12,26,27,28].

The needs of older orphans and their experiences have been neglected in comparison with those of orphans younger than 18 years. This is despite the existence of evidence that the negative effects of HIV orphan-hood do not get better as the subjects grow older than the age threshold of 18 years [2,24]. Older orphans who are the heads of their households bear a greater burden arising from the death of their parents from HIV-AIDS because they act as buffers for their younger siblings [24]. In addition, older orphans are often forced to drop out of school early. Several studies have associated early school leaving with psychological distress among orphans [17,28,29].

We explored how older orphans in YHHs experience and respond to maternal death and examined the strategies they employ to care for their younger siblings. It is evident that older orphans in YHHs need psychological support from their extended families, teachers, and communities in their assumed parenting roles. The study will suggest ways to address some of the problems that are facing older orphans in YHHs. 

## 2. Materials and Methods

### 2.1. Design and Setting of the Study

We used descriptive phenomenology to explore and understand how youths who are heads of their households respond to the death of their mothers. Consistent with phenomenological enquiry [30], only participants from whom we could learn and understand the loss of a mother were selected, using purposeful sampling. The participants were selected from households located in informal settlements of a township in the City of Tshwane, in South Africa. Non-profit organisations (NGOs) who provide material and psychosocial support to the orphans assisted the researchers to recruit orphans between 15 and 24 years old, who were the heads of their households and were taking care of siblings. Eighteen household heads were interviewed. The sample size is consistent with the phenomenological approach. According to Converse [31], a sample size of 1–10 participants is adequate in phenomenological enquiry to obtain information that is useful for understanding the phenomenon under investigation [32]. The design of the study, the population, and the setting are defined in detail elsewhere [33].

### 2.2. Data Collection

The second author (E.S.) conducted most of the in-depth interviews with the participants in their language of choice (Sepedi, IsiZulu, or IsiXhosa) using an interview schedule of open-ended questions. E.S. is trained in qualitative methods as well as conducting in-depth interviews as part of the master’s in public health program. The first author, an experienced qualitative researcher, supervised field work and conducted a few interviews in the beginning of the field work. Consistent with phenomenological interviewing [34], we asked four broad key questions using a semi-structured interview guide. Broad and open ended questions allow participants the opportunity to express their view point extensively [35]. In this study, this enabled the participants to respond freely about the loss of their mother and the strategies they used to cope. The participants were asked key questions about what it was like to be a young person heading a household, what it was like to lose a mother, and what strategies they employed to cope with the loss of their mother and care for their younger siblings. In addition, probes and follow-up questions followed the response of the participants, with a focus on how they experienced the loss of their mothers to get a comprehensive understanding of the phenomenon [36]. The participants were interviewed in their homes or at the premises of the NGOs. The interviews were conducted in private, were audio recorded with the permission of the participants, and lasted for 30–45 min. Written, informed consent was obtained before the interviews took place. Participants under the age of 18 years signed an assent form and social workers acted as their legal guardians and countersigned the consent forms.

In addition, we collected the participants’ sociodemographic variables including their age, gender, schooling status, number of siblings, access to social grants, type of housing, and employment status.

### 2.3. Data Analysis

The data analysis was inductive and followed Giorgi’s method of analysis summarised by Sundler et al. [37,38]. All the authors were involved in the data analysis under the guidance of the third author (S.M.), who has vast experience in qualitative research and data analysis, whereas the lead author (B.N.) already possessed skills in data analysis acquired through her doctoral candidacy. First, the transcripts were translated into English by the second author (E.S.), while listening to the recordings to ensure accuracy. BN also checked the translated interviews for accuracy. Thematic analysis in phenomenology begins with a search for meaning and patterns of meanings. The authors independently read a few of the transcripts repeatedly to develop familiarity with the data. Next, they searched for statements of meaning that pertain to the loss of a mother. They further explored how emergent patterns can be organized into themes that described the lived experiences of YHHs. The authors met regularly during the entire process to discuss and verify the accuracy of the emerging themes and the meaning of each theme. By moving back and forth between the emerging themes, the authors defined, named, and described the themes. The identified patterns of meaning were then organized into emerging themes after consensus was reached among the authors. The emergent themes were integrated and synthesised into meaningful wholes reflecting the phenomena experienced by the participants. NVivo (QSR International, Melbourne, Australia), a qualitative data analysis package, was utilised for the analytic process. Reflexivity was maintained by all the authors during the entire process; reflexivity eradicates any bias inherent in researcher beliefs and attitudes [39].

Rigour in qualitative research ensures that the findings of the study are credible and therefore transferable to other settings in similar contexts [40]. To ensure rigour in this study, all the authors analysed the data, thus ensuring that the interpretation was free of bias. In addition, they used peer-debriefing sessions, conducted the interviews in local languages, transcribed the responses verbatim in order to truly reflect the sentiments of the participants, and kept an audit trail to ensure rigour.

### 2.4. Ethical Considerations 

The protocol was approved by the Research and Ethics Committee of Sefako Makgatho Health Science University (SMUREC/H/53/2016: PG). Permission was also sought from the Department of Social Development and from the relevant NGOs. Participation was voluntary and the participants were told that they could withdraw from the interviews at any time if they wished to. Confidentiality was maintained and fictitious names were used to protect the identity of the participants. 

## 3. Findings 

### 3.1. Demographic Profile of the Participants

The sample consisted of 18 young people who were the heads of their households and caring for siblings, ranging from 1–4; most (10 out of 18) had one sibling. They were aged 15–24 years old and the average age was 21 years. Most (13 out of 18) were females and 5 were males. Most (12 out of 18) participants reported that they had dropped out of school and only 6 were attending school at the time of the interviews. Concerning the source of their income, 15 out of 18 reported that they had no income, 3 were receiving child support grants on behalf of their siblings, and 2 reported that their source of income was part time jobs. During the interviews, it was evident that the participants did not recall the time of death of their mother and some mentioned that they were very young when their mother died. They also reported that they were not told about the cause of death of their mothers (Table 1).

### 3.2. Emergent Themes

The phenomenological analysis resulted in the identification of three main themes: (1) the response to the parental death, (2) the consequences of the maternal death, and (3) the survival strategies employed by the heads of the households (Figure 1).

#### 3.2.1. Responses to the Parental Death

The data revealed emotions as a primary part of the participants’ responses to losing their mothers. Their narratives presented a variety of emotions related to their current situations that were grouped into three main sets of feelings: (1) sadness and loneliness, (2) anger and frustration, and (3) hopelessness. These sets of feelings were expressed towards their parents and extended family members, including their grandparents.

##### Sadness and Loneliness

Feelings of loneliness were expressed as a more common occurrence. The narratives of the participants revealed that they were alone and isolated from their peers and extended family members. There was a sense of alienation as they preferred to minimise their interaction with their peers and community members to avoid letting people know about their situations as well as the challenges they faced on a day-to-day basis.

“I cannot talk to anyone about my situations all the time. That’s why I prefer sitting alone and indoor. I feel like everyone sees what is going through my mind and want to know my situation.”(Keamogetswe, 24-year-old female)

“I don’t have friends at school. I stay home alone. The only friend that I have does not stay here, and we only talk over the phone.”(Kekalerato, 15-year-old female)

“I cannot keep friendship for long because I do not have time to socialise with them. During weekend when they go for parties, I could not because I have to look for casual job to bring food on the table for the kids. It is hard to be in a group that do not understand your situation. It’s better to keep to yourself.”(Onthatile, 20-year-old female)

The participants suffered from feelings of deep sadness due to the memory of their late parents and their inability to cope with the assumed adult role of caring for their siblings with minimal resources. The narratives revealed recollections about their dead parents and the desire to have them back. Their narratives suggested that they believed that if their parents, particularly their mothers, were still alive, their life situations would have been better. 

“The situation makes me sad. If my mom was still alive, I would have finished my degree and lived a better life than to work at the restaurant.”(Kephedile, 19-year-old female)

“After my father died, the family took him to Limpopo to bury him. We could not attend the funeral because we did not have money for transport. I was used to my father, we were so close, but I could not bury him. I am still hurting inside. It was so sad for me and my siblings.”(Tshepang, 25-year-old female)

“I feel sad because when everything happens I think of my mother. If she were still alive, she would do everything for me and guide me. It makes me feel sad because when I see those [relatives] buying other kids clothes and I am not included, I feel like I can dig a hole and get inside to hide. It hurts me a lot when I see other children receiving clothes and I do not. Sometimes it is hard because I would have thoughts of committing suicide so I follow my mother.”(Repholositswe, 23-year-old female)

##### Frustration and Anger

Many participants expressed feelings of anger that are often directed towards the mother for dying. Some directed their anger to the members of their maternal extended family. They felt frustrated about the situations they found themselves living in after the death of their parents and the ill-treatment they received from members of their extended family. 

“These people hate us… To think that they are relatives from my mother’s side… it makes me feel so hurt. I am angry at my mom. What is it that she did to these people that they hate us this much.”(Reyagoboka, 22-year-old female)

“I am so angry because if my mother was still alive I would not be faced with this situation. We never went to bed on an empty stomach when my mother was alive. My mother was not working but she used to manage everything well. I don’t like this situation. I get so frustrated.”(Keitumetse, 20-year-old male)

##### Hopelessness

The narratives revealed that the participants continued to yearn for their mothers long after their death. They expressed feelings of hopelessness about their situation. In general, there was a sense of lack of hope and a sense of powerlessness to influence their current living conditions in any positive way. The narratives suggest that they suppressed their emotions as a way of helping them to cope with the death of their mothers. 

“It makes me feel bad because there’s no one to look after us as we had our hopes and looked up to our grandmother to take us but she passed away two months ago.”(Repholositswe, 23-year-old female)

“I wish everything can be sorted and I live a better life. But, my mom is gone. She died and no miracles will happen. She won’t come back. I can’t keep on crying…, my mom is no more..., she is no more.”(Oratile, 25-year-old female)

“Sometimes I wish my mother was alive, but I need to be strong for these children [her siblings]. I can’t be always referring to situations when my mother was around. She is not here anymore.”(Keitlulwe, 24-year-old female)

“Eish, sometimes it breaks my heart. I feel like I can’t do it or I’m not strong enough and I feel like I’m failing or letting down the children’s mother because I can’t give them everything that they want. Sometimes I ask myself whether I am taking care of them the way their mother would have taken care of them.”(Omphile, 23-year-old male)

#### 3.2.2. The Consequences of Maternal Death

These included the loss of the family home and income, the loss of support from relatives, poor school outcomes, the denial of future aspirations, and social withdrawal.

##### Loss of the Family Home and Income

The study revealed that children in YHHs were living in absolute poverty and were extremely vulnerable and impoverished. They reported that they had been chased out of their parent’s homes by members of their extended families or stepfathers who took all that used to belong to their parents. 

“They forced us to move out of the main house immediately after we buried my mom, and we stayed in the back room. Then our stepfather kicked us out because he said he did not have kids and that he does not want any children in his house. They kicked us out during the night. We were forced to leave at night. We did not know where to go. I had to carry the kids, asking people for place to sleep.”(Reyagoboka, 22-year-old female)

“It’s tough because we can’t go and ask for help from my uncles because once my mother passed on they kicked us out and they said that she passed on without having a house.”(Kephedile, 19-year-old female)

In some cases, the participants were neglected and/or physically abused by their extended family prior to becoming heads of their households. When they experienced abuse or ill-treatment, they had no one to turn to for help. This led to them running away to live in youth-headed households. 

“After my mom passed on we used to live with my uncle, and at that place my younger brother was doing everything for them. He was doing the laundry, he was cleaning around the house, and I saw all of this, and that’s when I decided to get him and live with him. I would rather struggle with him.”(Lerato, 25-year-old female)

“When we go out with friends and we don’t come home before the curfew they beat us with a sjambok. To make matters worse they made us sleep without any food.”(Oampitsa, 19-year-old female)

The data revealed that even after the participants left the abusive home, they remained vulnerable because they did not have their own place. They stayed in rented shacks and lived in constant fear of being asked to leave by the owner. 

“We live in a shack that we are taking care of it for someone. When the owner comes back and says he wants his house back, where are we going to go?”(Reyagoboka, 22-year-old female)

“The owner of this place can take this place any time he wants, and when he does where are we going to go?”(Kelebogile, 25-year-old female)

Some of the participants reported that the child grants for their younger siblings were collected by members of the extended family, who used the money for their personal needs. 

“My sibling receives a grant but my uncle take it so it means I have to work.”(Lerato, 25-year-old female)

“My siblings are getting grants, but my aunt takes it. She does not give us anything. She supports her children and husband with that money.”(Keamogetswe, 24-year-old female)

##### Loss of Support from Relatives 

Most of the orphans did not receive any form of support from their relatives. As already mentioned, some were chased out of their parents’ homes while others were ill-treated and abused. 

“We have this aunt from my father’s side but she does not help us with anything. I feel hurt about that. There is another aunt from my mother’s side who comes occasionally but she also does not help us with anything.”(Tshepang, 25-year-old female)

“Yes, they are around but it’s not the type of relatives that will bother themselves by showing up. They do not come and visit, so I have that belief that we should not bother them too much.”(Tshepiso, 20-year-old male)

“Even if we ask help, they will not help us. If they kicked us out of the house do you think they will help us?”(Reyagoboka, 22-year-old female)

##### Forced to Socially Isolate

The HIV-related stigma and discrimination in the community led to withdrawal by the participants from the community. They cited incidents of labelling and name calling by the community who refer to them as AIDS orphans instead of addressing them by their names. Therefore, they see asking for help as bothering the next person and exposing one to shame or gossip. 

“The people in the neighbourhood talk too much. You will hear them saying orphans this, orphans that. That’s what I don’t like. I rather stay alone at home and sleep than to interact with them.”(Oampitsa, a 19-year-old female)

“The community will gossip about you. This is what I hate. I rather stay alone than mingle with the community, because one thing I know, after the death of my mom people kept talking. They said my mom this my mom that, my mom died of AIDS. When my mom died I was still young. I wouldn’t know what killed her. This is the type of life people who live here. They stay at their corners and gossip about other people’s life.”(Kephedile, 19-year-old female)

“I have pride, I will not talk to anybody about my issues, and they call us names and say orphans this orphans that and say orphans are tiring. We are not tiring. We need support.”(Kephedile, 19-year-old female)

#### 3.2.3. Survival Strategies Employed by YHHs

The narratives revealed that while the participants were vulnerable and lived in absolute poverty, they also exhibited resilience in dealing with their life circumstances. They used different strategies to address their day-to-day problems and challenges, such as performing casual jobs, accessing social grants, receiving assistance from community networks, and staying together with their siblings.

##### Dropping out of School to Perform Casual Jobs

Dropping out of school early is a strategy adopted by some of the participants. Most of the participants in the study were beyond the social grant plan because at 18 years old, the social grant is terminated. However, because they left school early without skills or a qualification, most remained unemployed and survived by performing part-time menial jobs in the community.

“If I am called to say there is a job for me, I make sure I leave everything to do that job, because I do not know when I will get another job. That is how we survive with my little brother.”(Kekalerato, 15-year-old male)

“I get part-time jobs here and there, but I feel like I can do something better with my life than to work at the restaurants and come home with swollen feet every day. But, what can I do, that’s the only money we have to survive.”(Lerato, 25-year-old female).

##### Ability to Prioritise Spending of the Social Grant

As most of the participants were too old to qualify for the social grant which provides financial relief to poverty-stricken children, the child support social grant for their younger siblings was their only source of income. However, they learnt to prioritise what to buy on a monthly basis.

“I hold the money on her behalf and make sure that we buy necessities in the house or if she is going on a school trip, I would give her lunch money.”(Kephedile, 26-year-old male)

“I use my brother’s money for buying food in the household, as I am not working. After buying the grocery I save R100.00 at the post office for him.”(Keitumetse, 20-year-old male)

“The children’s grant sometimes comes in handy in most cases, because we know how to budget the money.”(Oampitsa, 19-year-old female)

Some chose to stay with their siblings instead of living with members of their extended families so that they did not have to share the grant money with many people.

“At my granny’s place, no one is working. If my siblings stay there, I must split the little money I earn between here and there. If we stay together, a bag of mielie meal can last us for a month, but at my granny’s side, it cannot.”(Kephedile, 19-year-old female)

##### Assistance from Community Networks 

The narratives revealed that the community played an important support role for the participants after the death of the adult head of the family. They indicated that sometimes they received assistance from their neighbours in cash, in donated food, in donated clothing, and in assistance with school fees. Local NGOs provide their siblings with meals after school, assist with homework, and in some cases, provide school uniforms and donate food parcels.

“There are some people that support us in the community. Just like the woman next door, she supports us and she buys the children clothes and we never asked her to do that but she does it anyway.”(Oampitsa, 19-year-old female)

“Some of the neighbours support us. Our neighbour is a friendly person. When she buy groceries, she shares with us and would give us clothes that she received from her employer. Sometimes she would call me to come take some food for us to eat.”(Keitumetse, 20-year-old female)

Some of the participants revealed that some of their teachers were there for them after the death of their parents and encouraged them to continue with their schooling.

“At school, they are supportive. When my grandma died, the teachers did their level best to give me support. Some of the teachers offered to take me to matric dance but I refused. How could I go for matric dance when there was no food at home?”(Kephedile, 19-year-old female)

##### Staying Together in Youth-Headed Households

The participants felt strongly about staying together with their siblings despite the hardships that their households faced on a daily basis. They indicated that they are strong and happy together, but also that their experiences of living with their extended families after the death of their parents were characterised by abuse and maltreatment.

“I feel happy when I am around my siblings and I do not see ourselves living apart. We are facing challenges together and have a strong bond. If they had asked to take and raise my siblings immediately after my mother’s death, I would have agreed, because I had so much fear of how am I going to raise my siblings and step up in the family.”(Kephedile, 26-year-old male)

“I want us to live as a family and grow up together as one. Immediately after the funeral it seemed like they wanted to separate us, I think I would not have allowed them to take the kids away.”(Omphile, 23-year-old male)

“My young sibling and I have been through a lot for me just to give him up now. I struggled a lot to have the things we have now and I will not agree that they [her relatives] take him because I know how they will treat him and I do not want him to struggle the same way I struggled.”(Reyagoboka, 22-year-old female)

## 4. Discussion

This study has explored how older orphans who are household heads respond to maternal death and described the strategies that they employed to care for their younger siblings. The study has observed that the death of their parents was accompanied by multiple losses, such as loss of the family home, loss of multiple family members, loss of support from relatives, loss of friendships, loss of educational opportunities and future aspirations, and loss of childhood. This observation is consistent with previous research conducted with orphans in similar households, which concluded that these losses contributed to their psychological vulnerability [24,28,41,42].

Consistent with past studies, we found that when children lose their parents, they experience various psychological disorders. The study revealed that the participants had experienced psychological problems such anxieties, sadness, emotional pain, fear, loneliness, emotional shock, helplessness, low self-esteem, and yearning for their parents. Similar findings have been reported elsewhere [2,43,44]. Researchers have reported various factors that predispose orphans in child- or youth-headed households to long-term psychological problems. The most significant factor is that they often do not get the opportunity to grieve the loss of their parents [17,28].

We found that the participants could not mourn for their parents as some were chased out of their homes soon after the death of their parents, whereas others could not bury their parents and get closure. Research shows that in many instances, orphans mourn on their own and have no one to share their grief with [45,46]. Their having to grieve alone is accountable for the sense of helplessness and hopelessness that characterised orphans in these households. Consistent with previous research, the immediate role adjustment from that of a child to that of an adult after the death of a parent led to the expression of anger as a substitute for grief [13,47,48]. Their narratives revealed deep-rooted and persistent anger towards their mother for dying and leaving them in the condition they found themselves in.

Sadness was triggered by the loss of support from their relatives after the death of their parents. Instead of receiving support, the participants and their siblings were kicked out of the only home they knew. We found that not having adult guidance to head a household and deal with daily life situations aggravated their feelings of helplessness and hopelessness. This explains the observation by several researchers that household heads are more adversely affected because taking on the parental responsibilities comes at a cost to their own psychosocial well-being [10,41,49]. Consistent with previous research [45], the participants had not been prepared for taking on an adult role and were given no support or counselling to help them recover from their parents’ death. The lack of professional counselling and psychological support led to their experiencing unresolved grief and long-term psychological problems. Several researchers have reported similar observations [11,12,50,51]. The provision of continuous grief counselling services to orphans in child- and youth-headed households at the community and school level is critical to mitigate the impact of possible psychological disorders [41,45,52].

The loss of a parent results in a series of bereavements for children [3]. We found that the participants lived with the constant memory of their dead mother. They yearned continuously but silently for their dead mothers when they faced daily hardships such as the declining emotional and financial support from their relatives. As such, they believed that if their mothers had still been alive, their precarious situations might have been better. Moreover, the participants suppressed the emotional impact of parental loss to overcome their grief [24,28,53]. Persistent yearning for a mother has been reported elsewhere as a common expression of prolonged bereavement among orphans [28,43,54]. Popoola and Mchunu [24] caution that the suppression of the emotional impact of maternal death might compromise recovery from the loss and further render the household head vulnerable to psychosocial and emotional problems.

Consistent with past studies, the premature adoption of the parental role with no support and minimal resources [21,41] impelled the participants to employ strategies to care for their younger siblings. In the current study, as in others, it was established that the assistance and support given by community networks was not sufficient, was imbalanced, and lacked a cohesive approach and that the funding provided was not sustainable [12,41,50]. We found that most of the participants had dropped out of school to find work to provide for their siblings. Similarly, other researchers have reported on early school leaving as a strategy that is used by orphans who have parental responsibilities [5,8,28].

Whereas leaving school early was perceived as a positive sacrifice for their siblings, it had a negative effect on their psychosocial wellbeing and the framing of their future goals. Dropping out of school became a source of major emotional pain and psychosocial distress for them because of the trade-off between attending school and having enough food for their siblings. They lived in the realisation that they could never continue with their studies to fulfill their dreams of attaining a qualification [28,41]. Longing for the mother and wishing she was there was a common theme when they expressed the obvious loss of their dreams [28,41,50].

While researchers maintain that building resilience against problems is good, we found that some of the participants were using negative coping strategies such as self-neglect, the repression of emotions, and self-isolation to cope with the loss of their mothers and the day-to-day challenges of caring for their siblings. They self-sacrifice so that their siblings can get ahead and have a better life [24]. Gilbert and Charles [41] point out that the primary focus of household heads is their siblings rather than themselves and suggest that supports for youth in these households should not be child-focussed but should rather be family-focussed. Self-sacrifice is a negative coping strategy that resulted in the participants experiencing prolonged emotional distress.

We found high levels of feelings of loneliness as a result of the participants’ self-isolation from their friends and peers [21,28,55]. In our sample, many of the participants were not able to maintain friendships due to their low self-esteem. They used self-isolation as a means of avoiding explaining their precarious situation to others and then being subjected to gossip and rejection [28]. Context-specific interventions to prevent the development of negative coping strategies and to develop resilience should be implemented by organisations dealing with orphans in foster care and child- or youth-headed households. Being together with their siblings was a source of support for the participants, who when facing any life challenges, found comfort in the fact that they were together. They formed a strong bond through mourning their parents together and facing their hardship as a family [10,12].

### Limitations

The limitation of the study is that only the heads of households were interviewed and the effect of the maternal death on them could not be compared with the effect on their younger siblings.

## 5. Conclusions

This study highlighted the effects of maternal death on the psychological wellbeing of orphans who are heads of households. They did not get to mourn the loss of their parents and suffered from prolonged bereavement and repression of the emotional impact of the death of their parent. This psychological distress was most frequently manifested in prolonged pain and sadness, deep-rooted and persistent anger towards their mother for dying, anxieties, and despondency when confronted with the daily hardships experienced after the parent’s death. Continuous grief counselling should form an integral component of the psychosocial support services that are provided by NGOs to orphans soon after the death of a parent.

Dropping out of school was one of the main coping strategies used to augment the social grants, to help care for their siblings. Nevertheless, dropping out of school robbed them of their future goals. Overall, the participants used negative coping strategies to cope with the loss of their mothers and the day-to-day challenges of caring for their siblings.

Considering the connection between the maternal death and the loss of their family homes, the lack of material and financial resources, the absence of educational opportunities, the loss of their childhood, and the lack of support, it is important that psychosocial interventions be made available to support such orphans and respond to all their needs.

There is a need for the Government to strengthen the support networks in communities so that they can provide a cohesive, comprehensive, and sustainable response to the needs of young people living in child- and youth-headed households.

## Figures and Tables

**Figure 1 healthcare-08-00259-f001:**
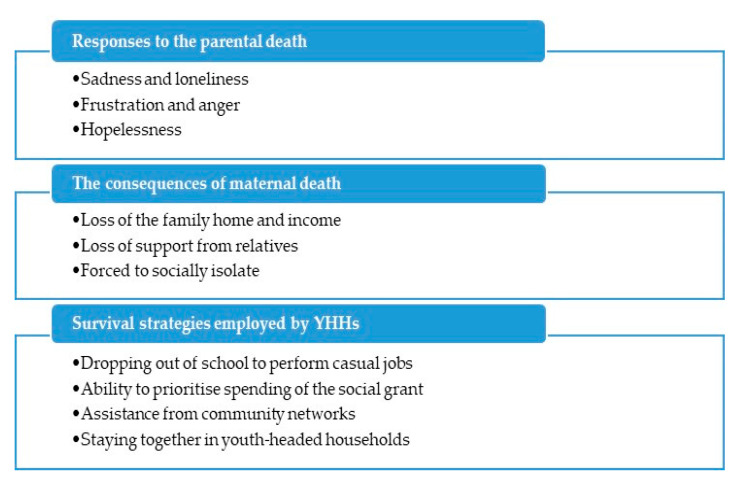
Emergent themes and subthemes.

**Table 1 healthcare-08-00259-t001:** Demographic profile of the study sample.

Variables	Categories	Frequency
Age category	15–20 years	5
	21–24 years	13
Gender	Female	13
	Male	5
Receive child grant	Yes	3
	No	15
Number of siblings	One	10
	Two	5
	Three	2
	Four	1
Dropped out of school	Yes	12
	No	6
Employment status	Part time employed	2
	Not employed	16

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
