# Peer review of "Responding to Maternal Loss: A Phenomenological Study of Older Orphans in Youth-Headed Households in Impoverished Areas of South Africa"

_healthcare, 2020, doi:10.3390/healthcare8030259_

Round 1
Reviewer 1 Report
It is appreciated that the authors have focused on the study of a situation in particularly vulnerable people such as older orphans in youth-headed households in impoverished areas of South Africa.
The methodology described is appropriate and consistent with the objective set. However, some aspects are not entirely clear, such as
- The recording of the interviews is not detailed if it was audio or video-audio.
- The previous experience or training of the interviewer is not clearly specified, nor is the experience in qualitative research of the researchers who participated in the analysis.
- Data analysis should be more detailed: units of meaning, topics, categories...or the groupings they made. It would be interesting to also include an explanatory table in the results.
There are incorrect expressions that confuse the reader, as in the Discussion "his study has explored the effects of.." In addition, a qualitative study with a phenomenological focus should be limited to describing experiences and experiences, but it does not seem to be the most appropriate way to identify "effects". It is possible that this is just an error of expression, but the authors should review the manuscript in this regard.
In the same sense, in the abstract, statements are made in the results that correspond better to a quantitative study than to a phenomenological type of study such as the one presented. The results of the abstract should be rewritten.
Finally, authors are advised to apply and attach the COREQ checklist along with the new version of the manuscript that includes the relevant modifications in relation to the suggested changes.
Author Response
Response to reviewer 1 comments.
We would like to thank the reviewers for their valuable comments. We have addressed all the comments as outlined below and in the relevant section of the manuscript. All corrections are highlighted in blue font colour
Point 1
It is appreciated that the authors have focused on the study of a situation in particularly vulnerable people such as older orphans in youth-headed households in impoverished areas of South Africa.
The methodology described is appropriate and consistent with the objective set. However, some aspects are not entirely clear, such as:
The recording of the interviews is not detailed if it was audio or video-audio.
Response
Details have been provided clarifying that a digital audio recorder was used to record the interviews (line 109).
Point 2
The previous experience or training of the interviewer is not clearly specified, nor is the experience in qualitative research of the researchers who participated in the analysis.
Response 2
The second author (ES) is trained in qualitative methods as well as conducting in-depth interviews as part of the masters in public health program. The first author, an experienced qualitative researcher supervised field work and conducted a few interviews in the beginning of field work (Line 97-99).
All the authors were involved in the data analysis under the guidance of the SM who has vast experience in qualitative research and data analysis whereas the lead author (BN) already possessed skills in data analysis acquired through her doctoral candidacy (Line 118-120). She co-facilitate seminars on qualitative data analysis for masters and doctoral students in the Department of Public Health. The second author received training in qualitative research and data analysis through the program. The third author, a professor of public health, directs the qualitative research program for the Department of Public Health and is responsible for training qualitative data analysis for masters, doctoral, and other academics in the university.
Point 3
Data analysis should be more detailed: units of meaning, topics, categories...or the groupings they made. It would be interesting to also include an explanatory table in the results
Response
We added details of the procedures followed for data analysis (page 117-135) and figure presentation of the themes
Point 4
There are incorrect expressions that confuse the reader, as in the discussion "this study has explored the effects of…..
In addition, a qualitative study with a phenomenological focus should be limited to describing and experiences, but it does not seem to be the most appropriate way to identify "effects". It is possible that this is just an error of expression, but the authors should review the manuscript in this regard.
Thanks for pointing out this error, the literature that supports this study refers to effects of maternal death because not all the studies were conducted using descriptive phenomenology as a method. However, for our theme and in the discussion we have rephrased effects of maternal death with consequences of maternal death. The manuscript has not expressed an aim to identify the effects but there is a thin line between consequences and effects. The themes emerged from the data in the true tradition of phenomenology but when the participants described how losing their mothers affected them, the theme effects emerged.
Point 6
In the same sense, in the abstract, statements are made in the results that correspond better to a quantitative study than to a phenomenological type of study such as the one presented. The results of the abstract should be rewritten.
Response 6
Thanks for pointing out this important aspect of writing out the results for phenomenological inquiry, we have rewritten the abstract in this regard.
Reviewer 2 Report
I found this paper interesting for its aims, quite new and unexplored area. However, I found several methodological and discussion concerns that have to be addressed and ameliorated.
Materials and methods
I think that the participants’ characteristics should be add in the text for example in a Table. You put demographic information in the results. I think that it should be more appropriate in the method section.
More details about how the interview was built are necessary to understand better the study design and the aims/methodology adopted. How the questions were made? Were based on the literature? How they were analysed? This part is not clear and should be explained better to appreciate more the scientific soundness of this manuscript.
In the sociodemographic information did you collect the mother’s death reasons or the age of the child when the mother died? How many time has passed from the mothers’ death? I think that this information could be interesting to have more idea on orphan’s emotional and psychological well-being.
Did you apply the inter-rater agreement between judges? What theory leads your interviews and the thematic analysis?
I suggest to have the interview vignettes adopting the italic format.
Are there some qualitative differences in interviews’ contents along orphans’ age or gender? Is it possible to evidence it? If not, put this consideration in the discussion section as recommendation for future research or study limit.
Possible clinical interventions could be added to the conclusion or to the discussion to give some suggestions about how move to implement orphans’ psychological condition.
Author Response
Response to reviewer 2 comments
We would like to thank the reviewers for their valuable comments. We have addressed all the comments as outlined below and in the relevant section of the manuscript. All corrections are highlighted in blue font colour
I found this paper interesting for its aims, quite new and unexplored area. However, I found several methodological and discussion concerns that have to be addressed and ameliorated.
Point 1 Materials and methods
I think that the participants’ characteristics should be add in the text for example in a Table. You put demographic information in the results. I think that it should be more appropriate in the method section.
Response
We added a table in the results section on the demographic profile of the study participants (page 4)
Point 2
More details about how the interview was built are necessary to understand better the study design and the aims/methodology adopted. How the questions were made? Were based on the literature? How they were analysed? This part is not clear and should be explained better to appreciate more the scientific soundness of this manuscript.
Response
The development of questions were in line with phenomenological approach whereby the investigator asks broad open ended questions to understand how the participants experience the phenomenon being investigated. In phenomenological inquiry, the questions are broad and open ended so that the respondents have sufficient opportunity to express their viewpoints extensively Giorgi (1997). The aim is to describe the participants’ experience in the way he or she experiences it, and not from some theoretical standpoint (Line 99-102)
Point 3
Sociodemographic information did you collect the mother’s death reasons or the age of the child when the mother died? How many time has passed from the mothers’ death? I think that this information could be interesting to have more idea on orphan’s emotional and psychological well-being.
Response 3
We added a table in the results section on the demographic profile of the study participants on the few quantitative variables that we collected. Most of the children did not recall the exact date when their mother died and we therefore cannot report on this one, some mentioned that they were very young when the mother died (page 4).
Point 4
Did you apply the inter-rater agreement between judges? What theory leads your interviews and the thematic analysis?
Response 4
We analysed the data using Giorgi’s five step method of descriptive phenomenological analysis. We did not apply the inter-rater agreement between coders. But we met regularly during the entire process of data analysis to discuss and verify the accuracy of the emerging themes and the meaning of each theme. We defined, named, described, and finalised the themes after we had reached consensus. We also maintained reflexivity to eradicate any researcher inherent bias, believes and attitudes (line 117-135).
Point 5
I suggest to have the interview vignettes adopting the italic format
Response 5
When we submitted the manuscript, we presented the quotation in italic format, but the manuscript was formatted by the journal to fit their style.
Point 6
Are there some qualitative differences in interviews’ contents along orphans’ age or gender? Is it possible to evidence it? If not, put this consideration in the discussion section as recommendation for future research or study limit.
Response 5
We did not analyse data using comparisons between ages and gender, we have added this as a study limitation
Point 7
Possible clinical interventions could be added to the conclusion or to the discussion to give some suggestions about how move to implement orphans’ psychological condition.
Response
Point taken and this has been added in the conclusion (line 7467-470)
Round 2
Reviewer 1 Report
The manuscript has improved considerably with the changes made by the authors in relation to the reviewers' comments. The current version if appropriate for publication in this journal.